# Olfactory Choice for Decomposition Stage in the Burying Beetle *Nicrophorus vespilloides*: Preference or Aversion?

**DOI:** 10.3390/insects12010011

**Published:** 2020-12-26

**Authors:** Pablo J. Delclos, Tammy L. Bouldin, Jeffery K. Tomberlin

**Affiliations:** 1Department of Biology & Biochemistry, University of Houston, Houston, TX 77004, USA; 2Department of Entomology, Texas A&M University, College Station, TX 77843, USA; tammylees1996@gmail.com (T.L.B.); jktomberlin@tamu.edu (J.K.T.)

**Keywords:** olfaction, sensory valence, *Nicrophorus vespilloides*, preference, carrion ecology

## Abstract

**Simple Summary:**

In the burying beetle *Nicrophorus vespilloides*, mating pairs raise their offspring together on a small carrion resource. We tested whether carcass age affected brood quantity and quality and found that pairs had significantly more offspring on fresher carcasses. To determine whether this reproductive benefit translates to an olfactory preference or aversion for carcass age, we conducted a series of olfactory trials testing adult mated female preferences for carcasses differing in age. Mated females spent more time associating with fresh carcass odors relative to those of an aged one, but also spent more time in empty chambers than those with an aged carcass, suggesting that mated females have a general aversion to those odors. Lastly, we characterized the odor profiles of fresh and aged carcasses to determine which compounds might be driving this olfactory aversion in mated female burying beetles.

**Abstract:**

Sensory cues predicting resource quality are drivers of key animal behaviors such as preference or aversion. Despite the abundance of behavioral choice studies across the animal kingdom, relatively few studies have tested whether these decisions are driven by preference for one choice or aversion to another. In the burying beetle *Nicrophorus vespilloides*, adult pairs exhibit parental care to raise their offspring on a small carrion resource. We tested whether carrion decomposition stage affected brood quantity and quality and found that mating pairs had significantly more offspring on fresher carcasses. To determine whether this observed reproductive benefit correlates with maternal preference behavior, we conducted a series of olfactory trials testing mated female preferences for mouse carcasses of differing decomposition stages. When given the option between fresh and older carcasses, females associated significantly more with fresher, 1-day old carcasses. However, this behavior may be driven by aversion, as females that were given a choice between the 7-day old carcass and a blank control spent significantly more time in the control chamber. We characterized volatile organic compound profiles of both carcass types, highlighting unique compounds that may serve as public information (*sensu lato*) conveying resource quality information to gravid beetles.

## 1. Introduction

Across animal systems, individuals rely on the sensory periphery to detect cues and signals emitted from a resource of interest or its microbial community and respond accordingly to them. Sensory processes can be divided into two main components: (1) a sensory detection phase, where a cue or signal binds to and activates a given receptor at the sensory periphery, and (2) a sensory processing phase which encompasses most of the downstream effects that this receptor activation has, which occur largely within the central nervous system. While sensory detection affects how well an individual detects a given cue or signal, or how much of a cue or signal is needed to elicit a response, sensory processing will largely affect what that response becomes from a physiological or behavioral view. Behavioral responses will typically be either positive or negative in response to the given cue or signal, which is referred to as valence. Sensory valence can differ not only among individuals, but within individuals as well, for example in response to reproductive state [1,2,3]. The identification of these context-dependent behavioral responses can result in the creation of model systems aiming to characterize the neural and behavioral mechanisms of sensory valence.

We aimed to advance our understanding of preference behavior in the burying beetle *Nicrophorus vespilloides*, Herbst, (Coleoptera: Silphidae) in order to determine its potential in serving as a system for studying sensory valence. *N. vespilloides* exhibit biparental care, where a male and female mating pair will often bury a small vertebrate carcass and use it as a brooding resource for their offspring [4,5]. Parents will often secrete antimicrobial exudates, eliminating certain harmful bacteria as well as seeding the carcass with bacteria from the beetle’s own microbiome, providing a more suitable feeding resource for their larvae [6,7]. This behavior suggests an important dynamic between the larval offspring of a mating pair and the microbial community associated with the carrion resource. Past studies provide further evidence of this interaction by showing differential responses in this system according to carcass age. For instance, research has shown colonizing and brooding on a relatively fresh carcass with little to no past insect or microbial colonization has significant fitness benefits for a mating pair’s offspring [8]. However, studies have also shown non-gravid females relatively prefer the volatile organic compounds (VOC) of a carcass that is further along in the post-bloated stages of decomposition [9]. In this case, though, beetles were responding to odors from large carrion sources (i.e., swine carcass), which might be indicative of food resources rather than brooding sites [5,10]. Regardless, field studies have shown burying beetles can be found on large cadavers in the highest abundances at the “post-bloating” and later decomposition stages [11,12], although the reproductive status of these individuals was not noted.

These contrasting results beg the question of whether gravid females experience a similar behavior as their non-gravid counterparts, or if they exhibit a relative change in sensory valence, as a relative preference for fresher carcasses would likely incur a reproductive benefit. In either case, these individuals are responding to public information (*sensu lato*) in the form of VOCs released from the carcass and its microbial community [13,14]. Certain burying beetles must compete with other insect species, such as blow flies [15] and other kingdoms, as the offspring compete with the microbial community for nutrients within the carrion resource [5,6,8]. However, the role of the reproductive state in processing this public information remains unclear.

To address these questions, we conducted olfactory behavior trials on mated female *N. vespilloides* to determine the relative preference for the odors of old and fresh carcasses. Furthermore, to determine whether this relative preference was driven by a preference for one odor or an aversion against another, we conducted one-choice assays testing gravid female preference for one carcass type relative to an empty chamber serving as a control. Furthermore, we reared offspring on relatively old or fresh carcasses to quantify the effects of carcass age on brood traits of interest. Lastly, we characterized the VOC profiles of relatively old and fresh carcasses to identify candidate compounds worth studying in the future to determine the chemical mechanisms that may drive the observed behavioral results obtained in this study.

## 2. Materials and Methods

### 2.1. Study System and Colony Maintenance

*Nicrophorus vespilloides* colonies were started from beetles received from Dr. Rebecca Kilner at University of Cambridge, UK. Beetles were kept under standard laboratory conditions (20 °C on a 16:8 light to dark cycle [16]) in individual boxes (5 × 5 × 5 cm) filled with moist garden compost and fed approximately 0.2 g minced beef twice per week. Mating was controlled by pairing individual males and females in plastic breeding boxes (15.25 × 7.5 × 7.5 cm) half-filled with moist garden compost and provided with a thawed mouse carcass (10–14 g, Layne Laboratories, Inc., Arroyo Grande, CA, USA).

Prior to preference trials, adult females (12–16 d post-eclosion) were exposed to adult male beetles for 4 h, which typically results in successful copulation and subsequent brooding in the lab. Preference trials were then conducted on gravid females the same day as mating, and females were returned to their boxes and allowed to lay on their provided carcass. Boxes were then checked daily to monitor larval development, and we confirmed that all females that had undergone preference trials were indeed gravid and successfully bore offspring. Upon the first signs of larvae, parents were removed to minimize effects of cannibalism on offspring measurements.

### 2.2. Effects of Decomposition Stage on Offspring Fitness

Fitness trials were run using the same treatments used in the olfactory choice trials (old and fresh carcasses). These individuals were paired and placed in boxes as with the rest of the colony. However, rather than being given a frozen mouse, they were either given a 1-day-old (fresh) or 7-day-old (old) mouse carcass. Adult beetles were removed when larvae began to migrate from the carcass to avoid cannibalism of offspring. Larvae were removed for measurement and were not replaced in the colony to avoid any compounding effects in future generations.

We measured brood characteristics from a total of 22 broods from two different generations reared on 22 carcasses (11 old and 11 fresh). Specifically, we counted the number of larvae produced on a given carcass, the total mass (in grams) of all larvae in a given brood, and the average mass per larva in a given brood.

We used a mixed effects analysis of variance (ANOVA) using the nlme package in R [17] to determine the effect of carcass age on brood traits. Generation was included as a random effect. We then used Tukey’s HSD (*p* < 0.05) to identify significant differences in the traits of broods raised on old vs. fresh mouse carcasses.

### 2.3. Preference Trials

Olfactory preferences of gravid females were assessed utilizing a design modified from [18] (Appendix A). Briefly, behavioral arenas were constructed from clear acrylic sheets (Lexan) and consisted of four chambers a focal female could travel between, and two chambers for housing olfactory models. At the start of a given trial, a female was placed into a 7 × 7 cm chamber, connected to a single 10 × 10 cm “neutral chamber”. From here, females could enter one of two 10 × 10 cm “choice chambers.” Each of these choice chambers is connected via a perforated wall to the chamber containing the olfactory model. Air flow was ensured via the attachment of a universal series bus (USB) powered fan to a perforated wall in the starting chamber, with air flow measured to be, on average, 0.6 ± 0.1 m/sec across the four arenas tested. Chambers housing olfactory models each contained a charcoal filter to allow sterilized air to flow from the outside, through the model chambers, and out the starting chamber, thus ensuring that focal females were presented with olfactory cues from both model chambers.

At the start of a given trial, females were placed into a given behavioral arena under red light, as in previous studies [19,20,21], thus ensuring that focal females’ behaviors were primarily driven by olfactory cues. For the next 20 min, their movement was video-recorded, and association times within each choice chamber were assessed, a commonly used and reliable proxy for preference in both the visual and olfactory modalities across animal systems [22,23,24,25,26]. We randomly selected the cue chamber to which a given carcass or control was assigned. If females did not visit either choice chamber during the 20-min trial period, they were deemed nonresponsive. All trials were conducted at approximately 21 °C between 1000 and 1300 h, which corresponds to between four and seven hours into the beetles’ dark cycle. To validate the assay design and method, and to test for side biases, we conducted a preliminary assay on 12 females (six of which were responsive) that were presented with two empty chambers. We found no evidence of side bias, as females did not significantly prefer one empty chamber over the other (Wilcoxon test: *p* = 1, Appendix A).

A total of 72 trials were conducted over three generations to account for potential temporal effects on female preferences. Each female underwent all three trials, with the order of trials and placement of olfactory models in each model chamber randomized. The effect of generation on female association time was assessed but was not significant (Kruskal-Wallis: *Χ*^2^ = 4.46, *p* = 0.11, df = 2). Therefore, behavioral results were pooled across generations. We used Wilcoxon signed-rank tests to determine whether females showed significant preferences for a given odor or control, depending on the comparison. Furthermore, we conducted odds ratio analysis using a generalized linear model examining the effect of preference trial (Old vs. Fresh, Fresh vs. Control, and Old vs. Control) on the proportion of responsive females in a given trial. Females were defined as responsive if they entered either choice chamber within the 20-min trial. Lastly, as an alternative estimate of preference, we noted which chambers mated females first entered during preference assays and conducted exact binomial tests to determine whether females preferentially entered a chamber containing fresh vs. aged carcasses [8].

### 2.4. Volatile Sampling and Analysis of Carrion Resource

To identify VOCs that might explain any observed female preferences for carcasses of varying age, the headspaces of a random subset of 1-day old and 7-day old mouse carcasses (*n* = 4 of each type) were sampled. Briefly, carcasses were removed from the freezer and individually placed into empty, sterilized containers that lacked soil for the specified time (1 or 7 days). The placement of carcasses into their containers was timed so that all carcasses would be sampled for VOCs at the same time. For VOC headspace sampling, carcasses were individually placed into pint-sized glass jars. Jars were then sealed with a layer of Parafilm, lids reattached, and sample volatiles allowed to accumulate within the headspace for 30 min to create a more concentrated sample. The jars were attached to the volatile pump system. Briefly, activated charcoal served to purify outside air entering the jar. Air then passes through the jar headspace and exits via the outflow containing a volatile trap packed with 30 mg HayeSep Q adsorbent (Volatile Assay Systems, Rennselaer, NJ, USA). Charcoal-filtered air was passed through the sample headspace at a rate of 1 L/min for 1 h, using a diaphragm pump (Parker Hannafin Corporation, Cleveland, OH, USA) system developed in the laboratory, and the concentrated volatiles were collected onto the adsorbent.

After sampling, volatile compounds within the trap were eluted from the adsorbent with one wash of 150 µL of dichloromethane and treated with 5 µL of 80 ng/µL N-octane to serve as an internal standard. Samples were then stored at −20 °C until gas chromatography-mass spectrometry (GC-MS) analysis. Upon completion of volatile sampling, traps were cleaned with 3 mL dichloromethane (MilliporeSigma, St. Louis, MO, USA), wrapped in aluminum foil, and reused for subsequent volatile samplings.

### 2.5. Gas Chromatography-Mass Spectrometry and VOC Identification

All GC-MS measures were conducted in collaboration with the Geochemical Environmental Research Group (GERG) at TAMU. Briefly, headspace volatile samples were analyzed using the Hewlitt-Packard 7693 autosampler, Agilent 6890 gas chromatograph, and 5973 mass selective detector (Agilent Technologies, Stockport, UK). A capillary column was used with helium serving as the carrier gas. The GC oven parameters were: the oven was held at 40 °C for 3 min, ramped up to 60 °C at 10 °C/min, a second ramp up to 150 °C at 3 °C/min, and a third ramp up to 250 °C at 20 °C/min, then held at 250 °C for 5 min. Compounds were identified based on mass spectra of compounds from the NIST 14 MS library of mass spectral database (Palisade Corp., Newfield, NY, USA).

### 2.6. Statistical Analysis of VOC Profiles

The relative abundances of identified compounds were determined by comparing a given compound’s total response value to that of the internal standard, octane. Relative abundances were calculated for each sample, and differences in volatile profiles according to carcass age were determined via permutational ANOVA (PERMANOVA) using the adonis function of the vegan package in R. We also conducted indicator species analysis [27] to identify VOCs that were significantly associated with one carcass type. Indicator species analysis (ISA) was conducted using the multipatt function in the indicspecies package of R. Significance thresholds were set at *p* < 0.05 for both PERMANOVA and ISA. VOC profiles were then visualized using non-metric multidimensional scaling (NMDS) in R using the vegan package [28].

## 3. Results

### 3.1. Effect of Carcass Age on Brood Characteristics

We found a significant effect of carcass age on total brood size, both in terms of number of larvae sired (one-way ANOVA: *F*_1,15_ = 11.3, *p* = 4.3 × 10^−3^, Figure 1A) and the total mass of broods (one-way ANOVA: *F*_1,15_ = 13.7, *p* = 2.1 × 10^−3^, Figure 1B). However, this effect did not individually affect the average mass of larvae (one-way ANOVA: *F_1,15_* = 0.111, *p* = 0.744, Figure 1C). On average, mating pairs that laid eggs on the relatively fresh, 1-day-old carcasses sired more offspring than those that colonized 7-day-old mouse carcasses (Fresh: 20.5 ± 2.42 larvae, Old: 13.2 ± 2.32 larvae). However, carcass age does not appear to be a limiting factor on the development rate of individual larvae, as they were roughly the same mass regardless of carcass type (Fresh: 0.128 ± 0.012 g/larva, Old: 0.124 ± 0.006 g/larva).

### 3.2. Effect of Carcass Age on Female Olfactory Preference

We found a significant effect of carcass age on olfactory-driven behaviors, as gravid female burying beetles spent significantly more time associating with the odors of fresh mouse carcasses than with those of older carcasses in the two-choice preference trials (Fresh = 417.2 ± 71.3 s, Old = 122.3 ± 38.3 s; Wilcoxon signed-rank test: *W* = 44, *p* = 7.8 × 10^−3^, Figure 2A). However, when given the choice between the odors of a fresh mouse carcass and an empty control, we found no significant difference in association time between the two chambers (Fresh = 106.9 ± 69.1 s, control = 267.3 ± 72.2 s, *W* = 50, *p* = 0.59, Figure 2B). Similar to the two-choice test, we found that females spent significantly less time associating with the odors of old carcasses than with the empty control (Old = 146.5 ± 47.8 s, control = 431.7 ± 91.9 s, *W* = 17, *p* = 0.028, Figure 2C). When measuring preference as the first chamber a mated female visited, we found no significant difference across all pairwise comparisons (exact binomial test, all *p* > 0.3, Appendix A). Together, these results suggest that gravid female olfactory-related behaviors are driven not by a preference for ideal carcasses, but a relative aversion against suboptimal ones. Furthermore, we found that the presence of both carcasses significantly affected the overall responsiveness of gravid female burying beetles (Fisher’s exact test: *p* = 0.026), as 92% of females (22/24 responsive females) visited at least one of the choice chambers when presented with the option between fresh and old carcasses, while only 63% responded when given the option between a fresh carcass and control (15/24), and 58% responded when given the option between an old carcass and control (14/24 responsive females, Figure 3). Specifically, mated females were significantly more likely to respond if given a choice between the two carcass types than between one carcass type and the control.

### 3.3. Effect of Carcass Age on VOC Profile

We found a significant effect of carcass age on the volatile profiles of mouse carcasses (PERMANOVA: *F*_1,6_ = 3.97, *p* = 0.029). We detected the presence of several VOCs on old carcasses that were absent in relatively fresh ones. Specifically, old carcasses produced several compounds which were absent on fresh carcasses and found to be significantly associated with old carcasses based on ISA (see Appendix A for VOC abundances relative to the internal standard). These compounds included dimethyl sulfides [dimethyl disulfide (DMDS), dimethyl trisulfide (DMTS)], phenethyl alcohol (PA), 4-ethyl-phenol, and hexanal (all *p* < 0.05, Figure 4).

## 4. Discussion

### 4.1. Relatively Fresher Carrion Provides Fitness Benefits to Gravid Females

In this study, we verified previous research highlighting the importance of the decomposition stage on the development of offspring of gravid female burying beetles [7,8]. In one previous study, a suboptimal, older carcass resulted in smaller larval offspring, on average, but no difference in the total number of offspring produced relative to a fresh carcass [8]. However, here we find that gravid females produced approximately 1.5 times more larval offspring, but no difference in individual larval mass, when rearing on 1-day-old relative to 7-day-old mouse carcasses. One possible explanation for the differences in results between these two studies is that the total brood sizes differed, with females in our study producing 2–3 times more surviving offspring than in the previous study. Regardless, the general finding that fresher carcasses provide a fitness benefit to gravid females is consistent across studies.

In this study, we did not want to disturb parental care and offspring development by counting egg numbers. Therefore, we do not know whether this fitness benefit provided by fresh carrion is due to an increased investment by gravid females into her brood through increasing the number of eggs laid or due to increased survival from the egg to larval stages when reared on fresher carcasses. Alternatively, parents may cull more offspring on an inferior resource in order to increase the fitness of the remaining offspring. All are possible, as female burying beetles have been shown to alter reproductive investment depending on age and perceived risk of death [29]. On the other hand, the bacterial community of an aged carcass could be detrimental to offspring survival, particularly in the earliest developmental stages [30,31,32]. Regardless, these results highlight the importance of decomposition stage in the development of *N. vespilloides*, which may help to explain their roles in carrion ecology and succession dynamics.

### 4.2. Gravid Female Nicrophorus vespilloides Show Relative Aversion towards Older Carcasses

Olfactory behavior trials showed that gravid females given a choice between the odors of a fresh versus aged mouse carcass spent significantly more time, on average, associating with the odors of a fresh carcass. Interestingly, one-choice assays revealed that this behavior may actually be driven or reinforced by a relative aversion towards the suboptimal aged carcass, as females spent significantly more time in the control chamber when presented with only an aged carcass, while those presented with a fresh carcass did not significantly associate with either chamber. This suggests that, in nature, gravid females may rely on a searching strategy that emphasizes avoidance of suboptimal carrion rather than one that focuses on attraction towards an optimal resource. Future studies should examine how these preferences in gravid females change with age. We may expect this relative aversion to diminish with age in order to maximize the likelihood of laying eggs, similar to how mating preferences of females in other systems become less stringent with age or when nearing the end of a mating season [33,34,35].

These behavioral findings on gravid females are in stark contrast to previous findings in non-gravid females, which have been shown to relatively prefer the odors of more aged carcasses [9], although these tests were done on relatively larger carcasses. However, our results are in accordance with more recent research characterizing breeding female burying beetle behavior towards carcasses [36]. There are several likely mechanisms through which this change in preference may occur. For example, individuals likely exhibit different repertoires of odorant receptors depending on age [37] and reproductive state [38]. It is possible that females are simply attending to different specific VOCs based on the odorant receptors they currently have. Alternatively, physiological or neural changes brought on by a change in reproductive state may drive the behavioral discrepancies observed between non-gravid and gravid female *N. vespilloides*, as in certain other insect systems. For example, in the noctuid moth *Agrotis ipsilon* (Lepidoptera: Noctuidae), mating inhibits male response to pheromonal cues, but not to odors unrelated to mating [39,40]. Likewise, female parasitoid wasps of the species *Nasonia vitripennis* (Hymenoptera: Pteromalidae) fail to respond to male sex pheromones upon mating [41,42]. Female *N. vespilloides* may, then, be responding to the same VOCs, but changes in the downstream processing of these cues due to reproductive state may cause an inversion in the valence of their behavioral response. Future studies should focus on conducting a repeated measures experimental design in order to verify within-individual switches in preference depending on reproductive stage. Nevertheless, our results together with past literature suggest that *N. vespilloides* may serve as a promising system for studying the neural mechanisms through which valence is assigned within individuals in the future.

It is generally assumed that optimal carcasses are rare and that burying beetles should, as a result, have a limited level of choosiness. However, our results show that, at least over a short time scale (20 min), gravid females will still exhibit aversive responses towards a brooding resource. It is possible that, given more time, females would ultimately select any carcass provided (indeed, our breeding experiment and other studies [8,31] confirm that females will breed on a carcass regardless of its decomposition stage). While previous studies largely describe parental choice behavior over a long period of hours or days, our results complement these studies by revealing how gravid females behave towards brooding resources during the initial stages of decision-making.

We found that the type of assay significantly affected responsiveness in gravid female *N. vespilloides*, as females were more likely to respond to the choice trial when presented with two carcasses rather than one carcass and a control. We believe that this is likely due to an increased concentration of odors reaching the individual at a given time, thereby making the assay itself more likely to elicit a response in the female.

Lastly, we should note that, in this study, we are making an assumption that the short-range behavior we are observing is a proxy for decision-making behavior made by gravid beetles. The general results from our behavior trials (a relative preference of gravid females for fresher over older carcasses) correlate with the results from our rearing trials (greater offspring production on fresher carcasses). However, it is possible that these behavior trials are not themselves predictive of decision-making that might occur over the extended distances over which these beetles can travel. Future experiments should examine how well short-range and long-range behavioral preferences correlate within this and other systems.

### 4.3. VOC Profile Analysis Reveals Candidate Volatiles Driving Avoidance Behavior

We found that the six-day difference in decomposition between our fresh and aged carcasses resulted in a significantly different VOC profile that females were able to respond differentially towards. Indicator species analysis revealed several candidate VOCs that should serve as promising compounds to further study to determine the chemical mechanisms through which gravid females are repelled by aged carcasses. Specifically, DMDS and DMTS are particularly promising compounds to further study. The abundance of DMTS increases with carrion age [43,44], and breeding female burying beetles of two other species (*Nicrophorus orbicollis* and *N. tomentosus*) are less attracted to fresh carcasses that have been dosed with DMTS [36]. *N. vespilloides* have previously been shown to be sensitive to these VOCs [45]. While the authors showed a relative preference by *N. vespilloides* for DMDS and DMTS in isolation, the reproductive states of the females were not documented. Furthermore, it is possible that it is a combination of VOCs that elicits this relative aversion in gravid females, or that behavioral responses are dose-dependent, as in other systems [46].

Relatively less is known about the roles that phenethyl alcohol, hexanal, and 4-ethyl phenol have on burying beetle behavior. Phenethyl alcohol could potentially have an important role driving gravid female avoidance of aged carcasses, as it has been shown to be an effective repellent in other invertebrate systems as well [47,48]. Interestingly, 4-ethyl phenol is an attractant and oviposition stimulant in other insect systems [49]. Hexanal, in isolation and in combination with other aldehydes, serves as an effective repellent in several invertebrate systems [50,51,52]. It is possible that these VOCs have an integrative effect on beetle behavior, with the prioritization of the detection of certain ratios of these compounds being largely dependent on the reproductive state. The effects of these compounds in attracting or repelling *N. vespilloides* should be further studied, as they could create new avenues of research aiming to determine the chemical mechanisms by which carrion colonization by *N. vespilloides* occurs. Furthermore, it should be noted that burying beetles must compete with other species, such as blow flies and other insects, for the use of a carrion resource [53,54]. These species must rely on the same bouquet of VOCs, in order to locate and colonize a given carcass. Future studies should examine whether competing species behaviorally respond to the same VOCs or rather to different constituent parts of the same public information in order to better determine the competitive dynamics involved in the colonization of a carrion resource.

## 5. Conclusions

The results from this study highlight the interaction between the volatile output of carrion microbial communities and *Nicrophorus vespilloides* in influencing both reproductive success and behavior. Specifically, we found that the potential fitness benefit incurred from colonizing a relatively fresh carcass may be driven not by a preference for fresh carcasses, but rather by an aversion towards suboptimal aged ones. Furthermore, we identified several VOCs that may be implicated in this observed aversion, and future studies should examine the roles of these compounds, in isolation and in combination, on female burying beetle behavior. These findings, in conjunction with past literature describing non-gravid female olfactory behaviors, suggest that *N. vespilloides* could serve as an ideal system for further studying the neural mechanisms driving changes in sensory valence.

## Figures and Tables

**Figure 1 insects-12-00011-f001:**
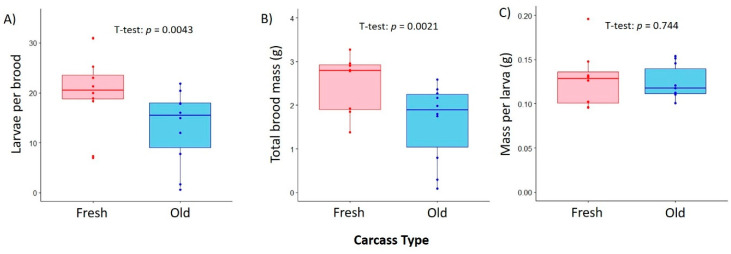
Box-and-whisker plots depicting the effect of carcass type on (**A**) total number of larvae sired by a mating pair of *Nicrophorus vespilloides*, (**B**) total brood mass, and (**C**) average mass per individual larva. Data points refer to individual samples, and boxplots denote median values and upper- and lower-quartiles.

**Figure 2 insects-12-00011-f002:**
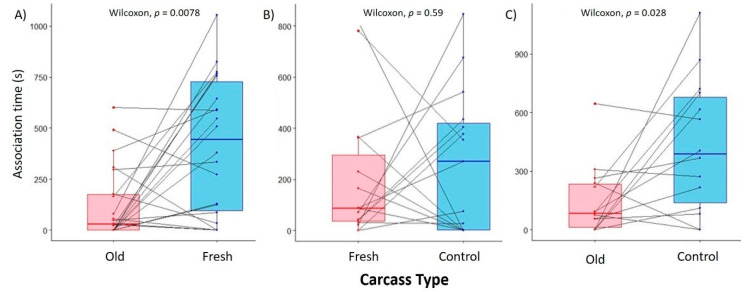
Box-and-whisker plots depicting mean female *Nicrophorus vespilloides* association times between choice chambers. (**A**) Old vs. Fresh carcass assay, (**B**) Fresh vs. Control assay, (**C**) Old vs. Control assay. Lines represent individual assays connecting the association times with both chambers for a given gravid female. Data points refer to association times of individual females, and boxplots denote median values and upper and lower quartiles.

**Figure 3 insects-12-00011-f003:**
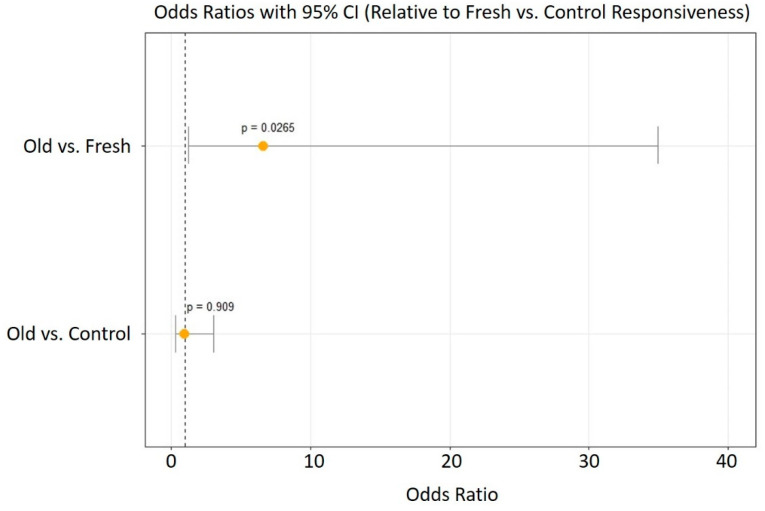
Odds of an individual gravid female *N. vespilloides* responding and showing a preference when presented with both an old and fresh carcass, a fresh carcass only, and an old carcass only. Odds ratios are presented relative to Fresh vs. Control assay (vertical line). Data points represent mean odds, and error bars represent the 95% confidence intervals.

**Figure 4 insects-12-00011-f004:**
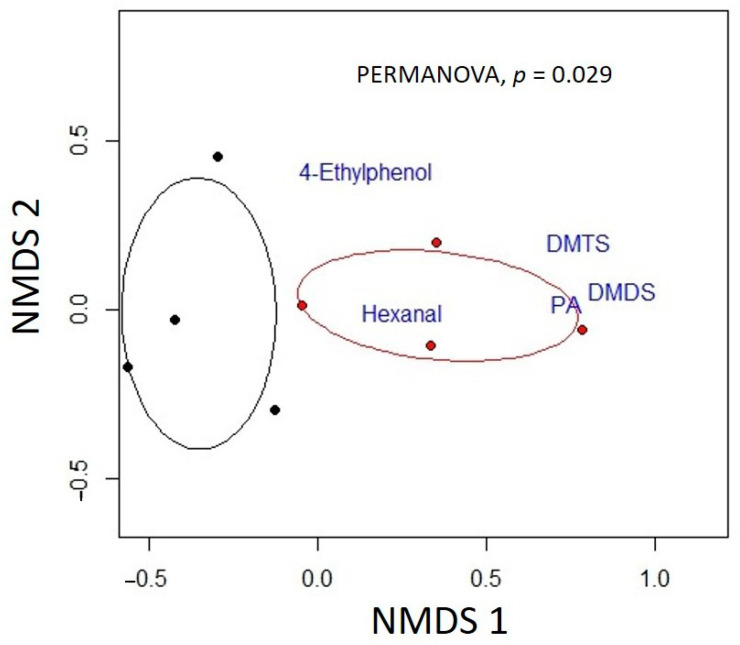
Nonmetric dimensional scaling plot describing differences in volatile organic compound profiles of fresh (black points) and old (red points) mouse carcasses used as brooding resources for *N. vespilloides* mating pairs. Data points denote individual carcasses, and ellipses depict the standard error around the mean centroid for each carcass age. Blue text denotes placement of volatile organic compounds (VOCs) significantly associated with aged carcasses (according to indicator species analysis) in multivariate space. The placement of a particular VOC in multivariate space was calculated in the vegan package in R as the means of each of the carcass non-metric multidimensional scaling (NMDS) scores weighted by the square root of that VOC’s abundance.

## Data Availability

All data files used for analyses described in this manuscript are available in the supplementary materials. Raw video files from olfactory preference trials are available from the authors upon request.

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
