# Peer review of "Olfactory Choice for Decomposition Stage in the Burying Beetle Nicrophorus vespilloides: Preference or Aversion?"

_insects, 2020, doi:10.3390/insects12010011_

Round 1

Reviewer 1 Report

The paper is well written and the study designed is appropriate.  There are just a few issues that should be dealt with to make it clearer and to explain some of the variation in responses.

Figure 2 there are some individuals that had no differential response or the opposite responses to the majority of females. Were all females used confirmed to be mated by some method like produced progeny after the trials?  If not then the possibility of some females not being mated should be mentioned.

Figure 4 is not clear.  The labels of compound space seem to be outside the confidence envelope for the aged carcass.  More explanation or a better way of representing the relative locations for the compounds is needed.  I would like to see a table or plot of the mean concentrations of the different compound identified as being given off by each carcass type included in the paper itself.

Since the microbial community is primarily responsible for the production of the VOC on the aged carcass, could the surroundings where the carcass is found also play a role in the relative aversion/attraction to the carcass?  How did the authors ensure that different carcasses would emit the same general VOCs? Could there be variation in the carcasses that resulted in part of the variation in behaviors observed?

Reviewer 2 Report

I have read manuscript # 1040473 (“Olfactory choice for decomposition stage …”). The writing is coherent, the experiments are understandable and the statistics reasonable. I agree with the overall assessment that this system could be a “good model for neural mechanisms during changes in sensory valence.” My biggest concern is the discussion of the relevant literature. A few important references have been missed and other cited work is not critically related to the findings in this work.

lines 262-266 (“more time… with the odors of a fresh carcass”). In the present study, females chose fresh over old carcasses. The initial choice in Rozen et al 2008 (cited but not in this context) was the opposite. In Rozen et al., the breeding-age beetles eventually settled on the younger carcass but not before most went first to the carcass with more odor. Is there an explanation for this difference in outcome?

lines 267-269 (“females may rely on … avoidance of suboptimal carrion”). This argument seems to assume that what the authors have investigated (walking toward a carcass that is close by) is a proxy for decision-making for a beetle flying over extended distances (beetles have been known to fly kms in one active period). Is this the assumption, and if so, why?

The lack of movement in the present study toward a fresh over a control chamber is also curious and leads to questions of the motivation of the beetles in the experimental apparatus. In fact, in all 3 setups the beetle moved to the area with the least sensory stimulation, although the difference was not significant for fresh versus control (were the beetles just trying to hide?). While the beetles are known to exhibit aversive behavior toward suboptimal resources, almost every beetle in previous studies used either a fresh or old carcass for breeding (in addition to Rozen et al., there is McLean et al. 2014 (JEB)). It is also generally thought that an adequate carcass is rare and ephemeral and that beetles should not be too choosy. Do the authors disagree with this generalization?

lines 274-275 (“behavioral findings are in stark contrast to previous findings in non-gravid females, which … prefer the odors of more aged carcasses”). A recent study with a different species of burying beetle made a similar contrast with the von Hoermann et al. 2013 study (Trumbo & Steiger 2020 Chemoecology).

lines 302-304. (“gravid females are repelled by aged carcasses, Specifically, DMDS and DMTS … promising compounds for further study”). Armstrong et al. 2016 (Heliyon), Kalinova et al. 2009 (cited in a different context) and Statheropoulous et al. 2011 (For Sci. Intl.) all found evidence that DMTS increases with carrion age (Armstrong et al. provide the description in the text, not the table). Trumbo & Steiger 2020 demonstrated that DMTS made a fresh carcass less attractive to breeders in the field.

lines 247-248 (“we verified previous research”). Rozen et al. 2008 also found that burying beetles had less reproductive success when breeding on older carcasses. They found, however, that with older carcasses there were the same number of larvae but of lesser mass, while the present study found the opposite. With their experimental design, they also had very small broods, which made interpretation problematic. The dataset here seems more typical for N. vespilloides (brood sizes). Are these differences worth discussing?

Minor points

line 128. A signal implies that the signaler benefits, on average, from sending the signal. This is not known for the carcass microbiota and is certainly not true of the carcass itself.

The authors use the term “foraging” (by gravid females) to mean searching for a breeding opportunity. This could be confusing to someone less familiar with burying beetles because nongravid just-emerged female (as in von Hoermann et al.) are only looking for a feeding resource while the focus of this study is on gravid females that are looking for a resource for breeding (though they may also feed).

Round 2

Reviewer 2 Report

The revision has addressed my concerns.

A few minor comments:

line 134. The experiment was done between 1000 and 1300. When was this relative to the manipulated light cycle (e.g., hours before lights off)?

line 166. Why is volatile in caps?

line 361. "colonizing" instead of "colonized"?

It could be made clear that carcasses were not placed on soil (if this is the case) prior to the headspace analysis.

Author Response

We would like to thank the reviewers for their comments on the current version of the manuscript. Please see the attached manuscript for revisions that have been made, and see below for point-by-point responses.

Response to Reviewer 2

Point 1: line 134. The experiment was done between 1000 and 1300. When was this relative to the manipulated light cycle (e.g., hours before lights off)?

Response 1: Preference trials were conducted between 1000 and 1300, which correspond to between hours 4 and 7 of the dark period of the beetles' cycle. 

Point 2: line 166. Why is volatile in caps?

Response 2: We thank the reviewer for catching this typo. We have corrected it (lowercase) in the manuscript.

Point 3: line 361. "colonizing" instead of "colonized"?

Response 3: "Colonizing" is the correct phrasing, and this line has been corrected accordingly.

Point 4: It could be made clear that carcasses were not placed on soil (if this is the case) prior to the headspace analysis.

Response 4: We agree that this could be clarified. As such, we have edited lines 155-156 to reflect that the sterile containers to which carcasses were added lacked any soil.